# The Panflute Technique: Novel 3D-Printed Patient Specific Instrumentation to Guide Curved Intra-Articular Osteotomies for Tibial Plateau Malunions

**DOI:** 10.3390/jcm13206175

**Published:** 2024-10-17

**Authors:** Nick Assink, Cornelia W. Binnekamp, Hugo C. van der Veen, Job N. Doornberg, Frank F. A. IJpma, Peter A. J. Pijpker

**Affiliations:** 1Department of Trauma Surgery, University of Groningen, University Medical Center Groningen, 9713 GZ Groningen, The Netherlands; j.n.doornberg@umcg.nl (J.N.D.); f.f.a.ijpma@umcg.nl (F.F.A.I.); 23D Lab, University of Groningen, University Medical Center Groningen, 9713 GZ Groningen, The Netherlands; c.w.binnekamp@umcg.nl (C.W.B.); p.a.j.pijpker@umcg.nl (P.A.J.P.); 3Department of Orthopedics, University of Groningen, University Medical Center Groningen, 9713 GZ Groningen, The Netherlands; h.c.van.der.veen@umcg.nl

**Keywords:** tibial plateau fracture, 3D, virtual planning, osteotomy, intra-articular, PSI, surgical guides, correction

## Abstract

**Background/Objectives**: 3D patient-specific corrective osteotomies are optimized for use with oscillating saws, thereby rendering it incapable of executing curved osteotomies. The aim of this technical note is to introduce and evaluate the Panflute technique, which facilitates curved osteotomies with precise depth control for intra-articular corrective osteotomies in posttraumatic tibial plateau malunions. **Methods**: A 33-year-old male patient with an intra-articular malunion was treated one year after index surgery of a lateral split-depression tibial plateau fracture with the Panflute technique. The guide design allowed for multiple drill trajectories in a curved path, recreating the original fracture lines. Cylindrical drill tubes in the guide were tailored to match bone trajectory length. This resulted in a patient-specific Panflute-like design enabling precise depth control, safeguarding posterior neurovascular structures. Secondly, the recreated fragment was reduced with a reduction guide, applied to the plate in situ, to facilitate reposition using the plate as tool and reference. **Results**: The procedure went without technical drawbacks or surgical complications. Postoperative assessment showed that repositioning of the osteotomized articular fragment was performed accurately: pre- to postoperative translational corrections were 5.4 to 0.5 mm posterior displacement for AP deformity (*x*-axis); 2.9 to 1.0 mm lateral to medial reduction (*y*-axis); and 5.9 to 0.6 mm cranial-caudal correction (*z*-axis). Clinically, at 3 months, the fracture united, the patient regained full flexion, and valgus defect-laxity resolved. **Conclusions**: The presented Panflute-osteotomy guide allows for a pre-planned curved osteotomy. Additionally, for every drill trajectory, the depth could be controlled. The proposed method may expand our surgical armamentarium of patient-specific 3D techniques and solutions for complex intra-articular osteotomies.

## 1. Introduction

Tibial plateau fractures may be complex fractures that usually consist of several intra-articular fragments and occur in specific patterns that guide treatment [1,2,3]. Surgical treatment of these fractures is often considered challenging due to associated fracture comminution, or patterns that require specific posterior fixation [4,5]. In more than 30% of the surgically treated tibial plateau fractures, a suboptimal operative result has been reported [6]. Malunions might lead to functional impairment, pain, and instability, potentially resulting in early-onset osteoarthritis of the knee joint, which may ultimately lead to the conversion to total knee arthroplasty. These malunions of tibial plateau fractures present significant challenges for the orthopedic trauma surgeon. Corrective osteotomy surgery provides a good option for these cases when carefully planned [7,8,9]. One recent study reported that revision surgery for residual displacement was required in 2% of the surgically treated tibial plateau fracture patients [10]. However, the maximum step-off and gap we may accept is subject of debate to date [11].

In the past years, an increasing number of methods for 3D-assisted corrective osteotomies have been introduced to correct for post-traumatic malunions [12,13,14,15]. Yet, these 3D-assisted methods mostly aim to correct extra-articular malunions. Corrective osteotomies for intra-articular malunions remain more challenging, and literature on 3D-assistance in these osteotomies is limited [16,17]. Current existing 3D technology for patient-specific corrective osteotomies is mostly optimized for use with oscillating saws, thereby rendering it incapable of executing curved osteotomies or facilitating precise depth control.

The aim of this technical note was to present and evaluate our novel technique for 3D-assisted intra-articular corrective osteotomies for post-traumatic malunions after a tibial plateau fracture, coined the Panflute technique. The technique is named because of the ‘Panflute-like’ design of the 3D-printed osteotomy guide that is introduced in this technical note. The concept facilitates the execution of curved osteotomies and precise depth control. This report describes the stepwise approach, encompassing 3D virtual planning, osteotomy- and reduction guide design, 3D-assisted surgery, and postoperative assessment.

## 2. Materials and Methods

A 33-year-old male patient, working as a personal trainer, was referred to our outpatient clinic with an intra-articular malunion one year after index surgical treatment of a lateral split-depression tibial plateau fracture (Schatzker 2; AO 41-B3) (Figure 1a,b). Patient presented knee instability, subjectively reported when weightbearing in flexion (squats, stairs, hiking), and resulting pain. Patient reported no instability in full extension while standing and showed very good quadriceps recovery after index procedure.

Physical examination showed lateral defect laxity beyond 30 degrees of knee flexion, with the medial collateral ligament complex intact. Patient identified resulting pain upon examination, which was recognized as the pain during his example of squatting. Plain radiographs of the affected knee suggested slight widening and distal translation of the split fragment and of a defect at the site of traumatic depression of the lateral tibial plateau (Figure 1c,d). It was felt that the malunited split fragment, with widening and distal translation, allowed for the lateral femoral condyle to sink in the bony defect and was explanatory for the patient’s complaints; it was theorized that the femur fell into the defect beyond 30 degrees of flexion.

### 2.1. Quantification and Segmentation of the Intra-Articular Malunion

A CT scan of both the affected as well as the contralateral intact tibia was performed. CT-data were then imported into the Mimics Medical software package (Version 25.0, Materialise, Leuven, Belgium). 3D reconstructions were made by performing a segmentation process using a bone threshold (Hounsfield unit ≥ 226) combined with the ‘region growing’ and ‘split mask’ functions in order to separate the tibial bone from the fibula and femur reconstruction (Figure 2a). The created 3D models of the tibia were subsequently imported into the 3-matic software (Version 17.0, Materialise, Leuven, Belgium). The contralateral tibia was then mirrored and aligned on the shaft of the affected proximal tibia to quantify the malunion in terms of a translational and rotational error. The mirrored contralateral side was subsequently used as a template to plan reconstruction (Figure 2b).

Quantitative 3D assessment of the CT scan (Q3DCT) revealed the following translational errors with respect to the contralateral side in three planes: (1) 5.4 mm in the anterior-posterior plane (*x*-axis); (2) 2.9 mm medial to lateral widening (*y*-axis); (3) and 5.9 mm cranial-caudal displacement. In addition, 3D rotational malalignment showed an error of 10.2° in the sagittal plane (ζ), 23° in the coronal view (θ), and 0.8° in the axial view (φ).

### 2.2. Virtual Surgical Planning (VSP)

Multiple drill trajectories were planned along the old fracture line resulting in a curved osteotomy cut that replicates the original split fracture (Figure 2c). Then, the lateral split fragment, which needed to be osteotomized and reduced, was virtually planned to its anatomical position by aligning it with the mirrored template (Figure 2d,e).

Surgical guides were then designed to translate the virtual plan to surgery. Guide design was also performed in the 3-Matic software. A total of three guides were designed: (1) the cutting; (2) the intermediate; and (3) the reposition guide. The cutting guide included our novel “Panflute” design, which facilitated the creation of a curved plane and precise depth control (Figure 3). The cutting guide, which included multiple drill trajectories to form the osteotomy plane, had two extensions to facilitate accurate positioning: an extension that covers the anterior tibial shaft (distal to the tibial tuberosity) and one that fits the in situ tibial plate.

Through this guide, three additional K-wires could be placed to facilitate the exact subsequent placement of the other two guides. The intermediate guide was designed to facilitate the process of predrilling the distal holes for the planned proximal tibia plate. To complete the process, we created a reposition guide to envelop the new proximal tibial plate to ensure correct positioning (Figure 4).

### 2.3. 3D-Assisted Surgery

The patient was positioned in the supine position. An anterolateral approach to the proximal tibia was used. Exposure included the lateral tibial plateau and plate, including a transverse capsulotomy and elevation of the lateral meniscus to allow for visualization of the articular surface. The osteotomy guide was positioned, and K-wires were placed to keep the guide in position and to serve as references for subsequent guides used at a later stage of the procedure. The position of the guide was verified on fluoroscopy, and after satisfactory positioning, a 1.8 mm drill was used through the respective holes in the Panflute drill guide to create the curved osteotomy plane (Figure 4a). An additional sleeve (Figure 4a, blue sleeve) was used to ensure the drill stopped at the planned depth. After all trajectories were drilled, the guide and implant were removed, and the planned bone fragment was separated from the rest of the bone with an osteotome (Figure 4b).

Subsequently, the small impacted depressed healed central fragment was identified, after opening “the book” by pushing the lateral split fragment open with a laminar spreader. The remainder of the small, depressed fragment was elevated and preliminary fixated with inside-out K-wires. The osteotomy was followed by predrilling the distal holes for the screws of the new plate using the intermediate guide, which was positioned with the in situ K-wires (Figure 4c). A new implant was then placed on the osteotomized fragment and attached with unicortical locking screws through the existing proximal screw holes from the old plate (Figure 4d). The reposition guide, which fitted on top of the new implant, was used to steer the fragment to the planned position relative to the bone (Figure 4e). After fixation of the plate, the reposition guide was removed and the wound was closed (Figure 4f). Figure 5 illustrates the per-operative use of the surgical guide. A supplementary animation illustrating the performed correction using the Panflute technical is provided (Appendix A).

### 2.4. Postoperative Assessment

A postoperative CT scan was made as part of the follow-up. Segmentation of the postoperative situation was performed in the same manner as the preoperative segmentation process. The virtually planned fragment reduction was then aligned with the postoperative model.

The preoperative, planned, and postoperative bone models were exported from the 3-matic software and imported within the Matlab (R2014B, Mathworks, Natick, Massachussetts, US) software. Translational error with respect to the planned positions was measured in terms of the anterior-posterior, left-right, and cranial-caudal distance (mm). The 3D rotational error (in degrees) was measured in coronal, sagittal, and axial directions.

## 3. Results

The procedure went without any technical drawbacks or intra-operative surgical complications. Clinically, the patient was satisfied and without pain at 3 months follow-up, regaining full knee flexion. Defect laxity, presented as a subjective feeling of instability resolved for squatting, climbing stairs, or hiking, such as experienced preoperatively. Postoperative radiograph showed quantified improvement of the height of the corrected lateral plateau, resulting in improved tibial alignment (Figure 6).

A quantitative 3D assessment of the postoperative CT scan was performed (Figure 7). The translational error with respect to the planned positions was subdivided into anterior-posterior, left-right, and cranial-caudal displacement parameters. An 0.5 mm posterior displacement (*x*-axes, initial 5.4 mm), 1.0 mm lateral displacement (*y*-axis, initial 2.9 mm), and 0.6 mm cranial displacement (*z*-axes, initial 5.9 mm) was observed. The 3D rotational error analysis in this case showed 3.8° in the sagittal view (ζ, initial 10.2°), 1.3° in the coronal view (θ, initial 23°), and 0.1° in the axial view (φ, initial 0.8°).

## 4. Discussion

Intra-articular corrective osteotomies of malunited tibial plateau fractures are technically demanding and require extensive preoperative planning [8,9]. 3D technology allows for preoperative planning and provides a tool for translating a virtual plan into the patient [15]. This technical note presents a novel 3D method using the Panflute concept in which the curved osteotomy cut is composed of several drill trajectories and therefore is not limited to a single plane. In addition, it allows for depth control of the osteotomy. This study shows that this method is feasible and allows for accurate reconstruction of the malunited bone.

Typically, osteotomies are performed with the use of a sawblade and osteotome to separate the malunited bone into the required bone fragments for reduction. For extra-articular osteotomies, usually, a plain cut is sufficient since secondary bone growth will not impair function. When one is treating an intra-articular malunion, plain cuts are often not sufficient. Particularly since you are dealing with cartilage, which should be cut and aligned anatomically to allow for improved functional outcome. Due to these difficulties, intra-articular corrective osteotomies remain challenging, and only a few studies report a limited number of patients who underwent conventional intra-articular corrective osteotomies [8,9,18]. Yet, without 3D guidance, accurate reduction of malunions may lead to unpredictable results [15].

3D-guided extra-articular corrective osteotomies are well described in literature; however, only a few studies describe 3D-guided intra-articular corrective osteotomy. Recently, a few case studies described corrections of malunited tibial plateau fractures with 3D assistance. Pagkalos et al. used a 3D-printed surgical guide through which the saw blade was steered to perform an osteotomy through the eminentia [19]. Two other recent studies reported the use of a surgical guide to place k-wires along which subsequently the saw blade was guided [16,20]. Our specific Panflute design differs from the beforementioned methods, enabling the creation of multiple drill trajectories that together form the curved osteotomy cut while controlling the depth for each hole. In addition, our method also incorporates a reposition guide to allow for accurate repositioning of the affected bone fragment.

The key aspect of this in-house workflow was the close collaboration between surgeons and technical physicians to ensure the best patient matched solution. However, throughout the planning and execution of the 3D-assisted correction according to our method, a few difficulties were observed. Firstly, the VA-LCP proximal tibia plate, which was already in situ resulted in significant scattering of the CT scan. Consequently, the design of the guides was complicated since the bone surface is affected by the scattering, which made the segmentation of the bone less reliable. Ideally, for accurate 3D planning, no hardware should be in situ. Yet, our case demonstrated a good solution by using the plate as a reference for the positioning of the osteotomy guide. The plate provided a stable reference point, enabling accurate positioning of the guide. Another complicating factor was the patellar tendon that blocked the positioning of the osteotomy guide, which is not accounted for on CT imaging. In our case, the guide was slightly too bulky, and part of the guide that interfered with the tendon had to be cut. As this was anticipated, the guide was planned with other landmarks for positioning, and it did not affect our intra-operative planning; soft tissues should be taken into account during the planning. This challenge also highlights one of the main limitations of using this type of surgical guide, as it often necessitates a wider incision. Additionally, the fact that the feasibility of the technique was only assessed in a single case represents another limitation. Although this article is intended as a technical note, further experience and validation with more cases are essential to optimize and refine the technique.

In conclusion, this technique proved to be accurate and resulted in anatomical reduction of the malunited fragment as compared to the contralateral templated side. The use of the Panflute concept for intra-articular osteotomy guides provides the possibility of curved cuts. In addition, from every drill trajectory, the depth could be controlled. The proposed method expands our surgical armamentarium of patient-specific 3D techniques and solutions for complex intra-articular osteotomies.

## Figures and Tables

**Figure 1 jcm-13-06175-f001:**
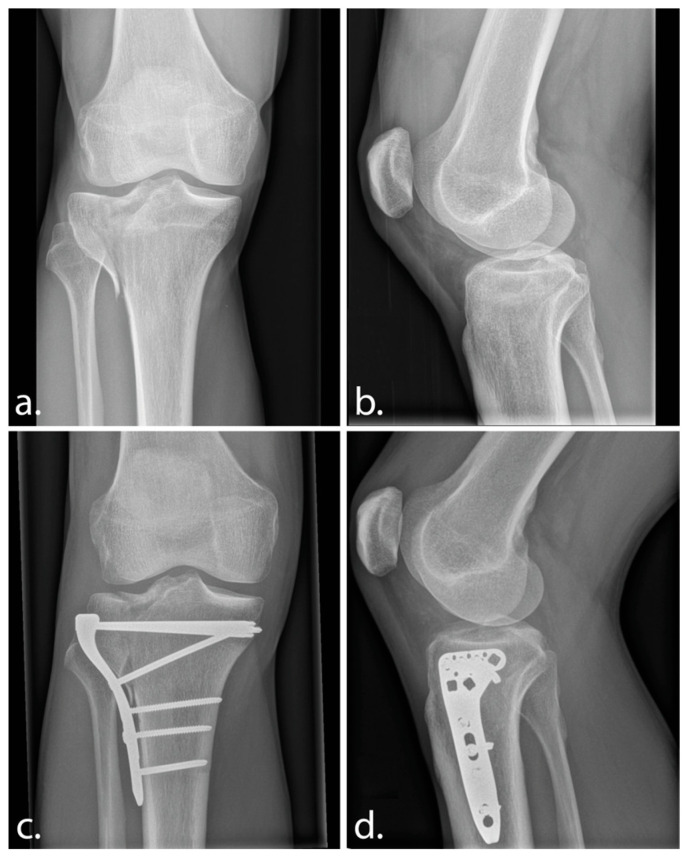
Anteroposterior and lateral radiograph of the initial tibial plateau fracture (**a**,**b**) and of the situation 1-year postoperative (**c**,**d**).

**Figure 2 jcm-13-06175-f002:**
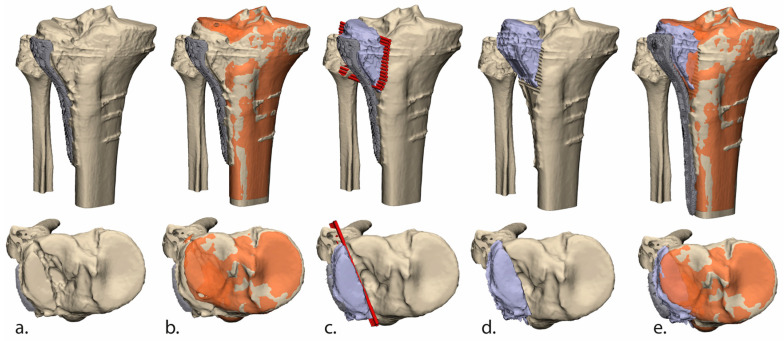
Process of 3D virtual surgical planning. (**a**) 3D reconstruction of the affected bone is created from the CT scan. (**b**) The contralateral unaffected side (orange) is mirrored and aligned to serve as a template for reduction. (**c**) The osteotomy cut (red) consisted of several drill trajectories and was planned to cut the malpositioned fragment. (**d**) The malpositioned fragment is virtually reduced to its anatomical position. (**e**) The new position is verified with the matching of the mirrored contralateral side.

**Figure 3 jcm-13-06175-f003:**
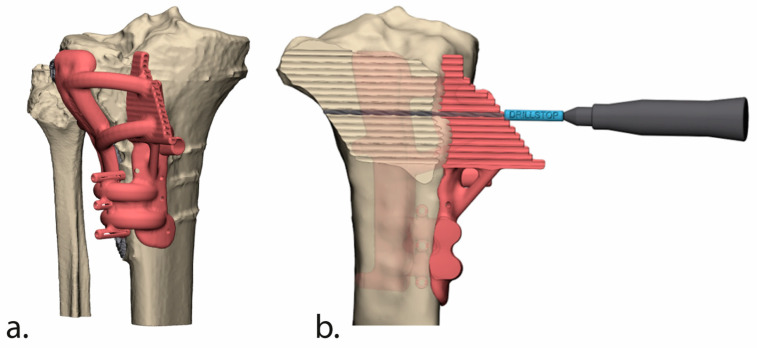
The ‘Panflute guide’ in anterior (**a**) and cross-sectional view (**b**). Tailoring the cylindrical drill tube to match the bone trajectory length for each hole creates a Panflute-like design that enables precise depth control and safeguards posterior vascular structures.

**Figure 4 jcm-13-06175-f004:**
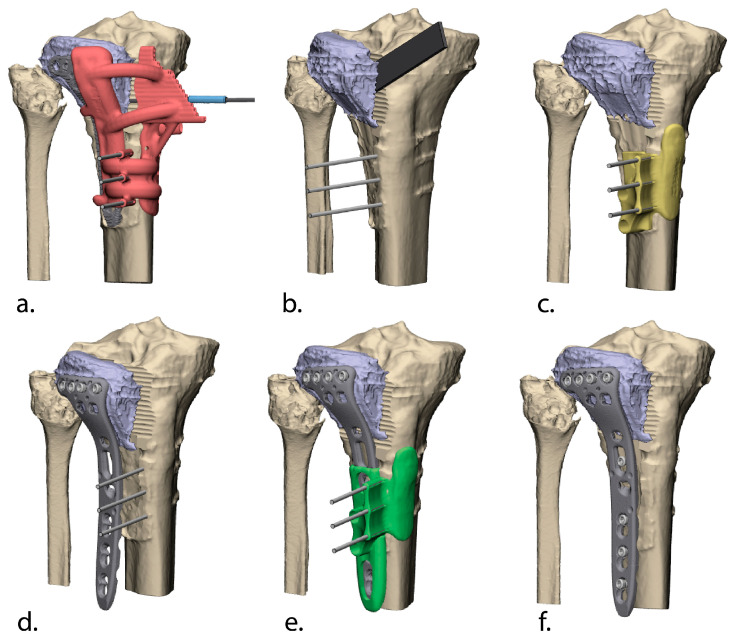
3D-assisted surgery: The Panflute osteotomy guide (red) was positioned on top of the in situ plate and kept in position with K-wires, which later serves as a reference for the intermediate (yellow) and reposition (green) guide at a later stage of the procedure. After satisfactory positioning, a 1.8 mm drill was used through the different holes in the Panflute guide to create the osteotomy plane (**a**); after all trajectories were drilled, the guide and implant were removed, and the bone fragment—that needs to be reduced—was separated from the rest of the proximal tibia with an osteotome (**b**); the osteotomy was followed by predrilling the distal holes for the screws of the new plate using the intermediate guide (**c**); then the new plate was fixated on the osteotomized fragment using the old proximal screw holes (**d**); the reposition guide, which fitted on top of the new implant, was then used to steer the fragment to the planned position relative to the bone, after which the plate was fixated on the shaft (**e**); after fixation of the plate, the reposition guide was removed, and the wound was closed (**f**).

**Figure 5 jcm-13-06175-f005:**
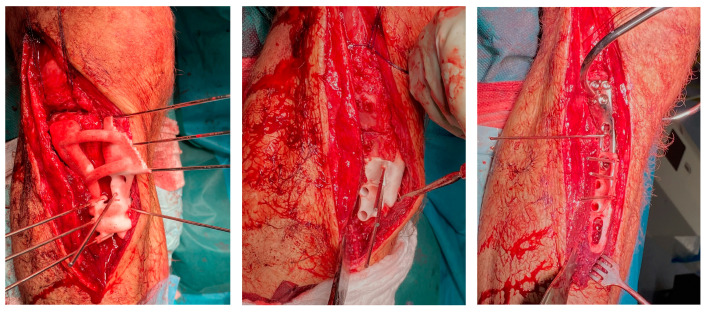
Per-operative usage of the different surgical guides: Panflute guide (**left**), intermediate guide (**middle**), and reposition guide (**right**).

**Figure 6 jcm-13-06175-f006:**
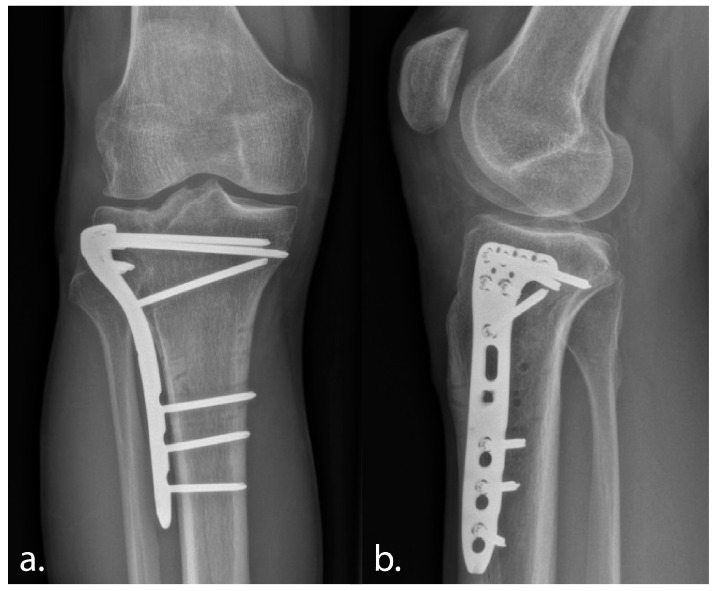
Postoperative anteroposterior (**a**) and lateral radiographs (**b**) at 3 months follow-up demonstrating improved alignment and progressive consolidation.

**Figure 7 jcm-13-06175-f007:**
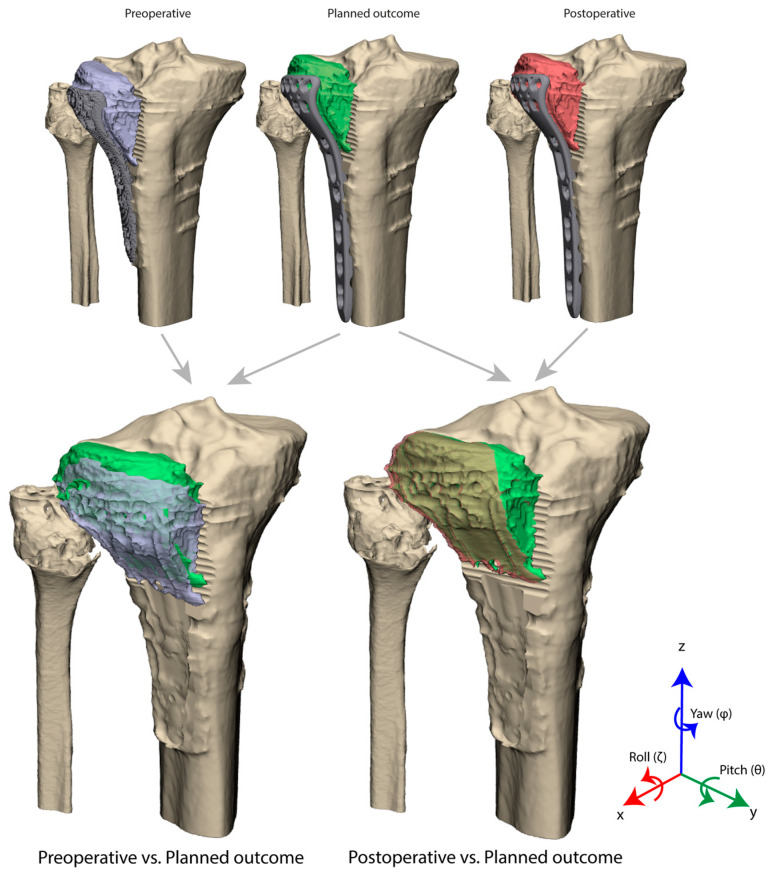
Preoperative (purple), planned (yellow), and postoperative (green) position of the reduced fragment. 3D assessment is performed to assess the fragment position before and after surgery as compared to the planned outcome. Difference is assessed in terms of translation (Δx, Δy, Δz) and rotation (Δζ, Δθ, Δφ) in three axes.

## Data Availability

Data are contained within the article and Appendix A.

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
