# Peer review of "The Panflute Technique: Novel 3D-Printed Patient Specific Instrumentation to Guide Curved Intra-Articular Osteotomies for Tibial Plateau Malunions"

_jcm, 2024, doi:10.3390/jcm13206175_

Round 1

Reviewer 1 Report

Comments and Suggestions for Authors

**Introduction** 

The purpose of this paper was well explained. However, providing a brief description of the complications and prognosis that can arise from malunion would engage readers more in the purpose of this paper. 

Please include informed consent from the patient and Institutional Review Board (IRB) approval for this case.

**Line 147-148** 

Were you able to accurately position the guide on the previous plate? Did any subjective surgical skills come into play during this process? 

Although actual surgical photos were provided, it would be helpful to also include pictures of the actual guides created using 3D techniques. 

The "points of caution" described below line 234 were well articulated. 

Lastly, I recommend considering the presentation of a series by gathering various cases. Thank you for introducing this excellent method.

Reviewer 2 Report

Comments and Suggestions for Authors

Dear Authors,

it is my pleasure to review your study.

Article titled "The Panflute Technique: Novel Three-Dimensional Printed Patient Specific Instrumentation to Guide Curved Intra-Articular Osteotomies for Tibial Plateau Malunions" raises an interesting topic but I have a few of doubts.

First of all, the article should be prepared in accordance with the journal's guidelines:

- The reference format should be corrected.

1.The aim of the article should be clearly stated both in the abstract and at the end of the introduction. It should be corrected. 

2.The AO classification was mentioned, but it is worth mentioning the Schatzker classification. And to determine the degree of fracture in the Schatzker classification of the discussed 33-year-old patient.

3.It would be worth presenting the results of the CT scan (not only 3-D) of the discussed 33-year-old patient before and after surgery. As we known, in this type of injury the gold standard of diagnostic imaging is CT, not X-ray.

4.It is worth adding the limitations of this study. 

5.The presented technique is based on a single case. Its implementation and dissemination requires more cases.

6.There is oo conclusion at the end of the article. It should be corrected.

The manuscript requires corrections, but is interesting and shows new possibilities in knee surgery.
